# The Needs of Parents of Children Suffering from Cancer—Continuation of Research

**DOI:** 10.3390/children9020144

**Published:** 2022-01-23

**Authors:** Anna Lewandowska

**Affiliations:** Institute of Healthcare, State School of Technology and Economics, Czarnieckiego Street 16, 37-500 Jaroslaw, Poland; am.lewandowska@poczta.fm; Tel.: +48-698-757-926

**Keywords:** cancer, children, parents, needs

## Abstract

Background: Parents experience many healthcare needs when caring for their sick children. Research shows that parents of oncological children have a high level of unmet needs, including psychosocial, emotional, physical, informational, financial, educational, and spiritual needs. To date, little quantitative research has been carried out on the specific needs of parents of children with cancer, which creates uncertainty about what areas should be addressed in care. This study investigated the prevalence of unmet needs among parents of children with cancer. Methods: A population survey was conducted between 2015 and 2020. Caregivers of children diagnosed with cancer were invited to participate in the study to assess their problems and needs. Results: The analysis found that 97% of the participants experienced some level of need for one or more items, and 73% of the respondents reported a moderate or high need for one or more items. In the field of medical information, 70% of respondents had moderate or high needs, 55% of parents reported a moderate or high level of need for help in the psychological or emotional field, and 30% in the financial domain. The prevalence of moderate or high need in the remaining domains ranged from 10% to 15%. Conclusions: Parents of children with cancer experience a high level of needs, especially psychological, emotional, and information. These data suggest that the existing healthcare system does not meet the needs of parents of sick children. The results show the need to investigate the mechanisms by which healthcare providers can use the healthcare system to identify and meet needs.

## 1. Introduction

A child’s chronic disease affects every sphere of their life—physical, mental, socio-economic, and behavioral, and because the illness is a central event for the family, it also affects the child’s closest family, especially their parents and guardians [1,2]. In the scientific literature, a family caregiver is defined as a person who has a significant emotional bond with the patient. The caregiver is a family member who is a part of the patient’s family life cycle, provides emotional, expressive, instrumental, and tangible support, and provides assistance and comprehensive care during a child’s chronic, acute disease or disability [3]. Childhood cancer is a chronic disease that requires continuous care through treatment, hospitalization, and coping with the side effects of therapy. It impacts the quality of life of the family caregiver on a personal, social, and professional level and increases their susceptibility to related emotional and physical disorders [1,2].

When faced with information about a child’s cancer diagnosis, families often experience shock, suffering, and changes in their world view and everyday family life. Parents quickly immerse themselves in the medical system while their lives are consumed by medical care and managing the needs of a sick child [4]. A sudden change in the life situation disturbs the psychosomatic balance and the functioning of the whole family. Psychophysical fitness deteriorates, somatic ailments increase, social and professional contacts are limited [5]. The crisis caused by the disease and hospitalization of children is one of the main sources of tension and anxiety for families, as the family is the primary source of support for sick children [6]. Parents have to cope with many challenges, such as side effects caused by treatment, life-threatening situations, experiencing the death of other patients, financial problems, an unstable professional situation, and emotional problems in themselves and other family members [7].

The effects and consequences experienced by families taking care of children with cancer include the risk of developing anxiety, depression, and parental stress, psychosocial distress, and burdening and burnout of the caregiver, often leading to a deterioration of mental well-being [8,9,10]. Caring for a child with cancer is an emotionally exhausting task, causing physical, mental, social, and economic well-being to decline as the disease progresses [11,12,13,14,15]. Parents experience many healthcare needs when caring for their sick children. Research shows that parents of oncological children have a high level of unmet needs, including psychosocial, emotional, physical, informational, financial, educational, and spiritual needs [16,17]. The assessment of the needs of families, especially at the beginning of the disease, is of great importance as it enables the prediction of psychosocial risk and potential stress. It also allows health workers to respond to their needs. It is very important to quickly identify and respond to the needs of families to minimize the negative effects of the stressor, allowing families to focus on patient care. Parents will always play a decisive role in patient care; they directly interact with sick children and deal with related problems and challenges. Therefore, we need to know what needs are urgent for the parents of sick children in order to provide them, together with the healthcare institution, with full support and help [18,19,20,21].

To date, little quantitative research has been carried out on the specific needs of parents of children with cancer, which creates uncertainty about what areas should be addressed in care. Therefore, the aim of this research is to identify the unmet needs among parents of children with cancer.

### Objective of the Work

The study aimed to assess the needs of parents of a child with cancer and to analyze the level of their dissatisfaction.

## 2. Materials and Methods

### 2.1. Study Design

The study was descriptive. A nonprobability sampling technique was used. The qualitative study was conducted in pediatric oncology and hematology departments of hospitals and oncology clinics in Podkarpackie Province in 2015–2020. Hospitals and clinics provided treatment and care by the public healthcare system to pediatric patients diagnosed with cancer, living in Podkarpackie Province. Parents of children diagnosed with cancer were invited to participate in the study to assess their needs. Due to the small size of the sample, the share of respondents with fairly consistent characteristics was important. The inclusion criteria for this study were as follows: family caregiver of a child with cancer over 18 years of age, provision of an informed consent form, confirmed diagnosis of childhood cancer in a child under the age of 18 without any previous chronic or life-threatening disease, and knowledge of the Polish language. All persons invited to participate provided their consent. The exclusion criterion was the diagnosis of neoplastic disease shorter than three months, because the initial period of diagnosis and treatment is associated with an enormous psychological burden and the need to adapt the patient, as well as the entire family, to the patient’s situation, which may introduce errors in the results. The study excluded parents in a poor physical and emotional state. Each invited person received information about the purpose of the study.

### 2.2. Participants

A cross-sectional study was conducted involving 800 family caregivers of children with cancer. Parents were invited to participate in the study during their children’s hospitalization. To participate in this study, participants had to be 18 years of age or older, be a father or mother caring for a child with cancer, and have read and signed an informed consent form before enrolling in the study. Potential participants who were illiterate or refused to volunteer were excluded from this study. Only one of the parents who performed the main care function (i.e., was present during a hospital stay or an inspection at the clinic) took part in the study. Each invited person was informed about the purpose of the study. After obtaining informed consent, subjects were asked to complete a questionnaire. The respondent was allowed to complete an online questionnaire or a paper version.

### 2.3. Research Procedures

The study was approved by the Bioethics Committee (Resolution No. 386/2009 and 4 December 2017). All participants were provided with information regarding the study’s objective and their research rights, particularly regarding the fact that there were no consequences if they decided not to participate. Participation in this study was voluntary. Prior to completion, participants were all informed of their rights as outlined by the Helsinki Declaration.

### 2.4. Method

#### Questionnaire

The method used in the study was an interview questionnaire. It was a standardized measuring instrument, verified within a month and assessed for internal consistency (receiving the Cronbach’s alpha coefficient of 0.83). Parents filled out a questionnaire that included open and closed questions in order to obtain epidemiological information and assess the areas of possible needs: psychological, medical, communication and information, everyday life, social, spiritual and financial needs. Parents were asked to indicate their level of needs on a 4-point Likert scale (1—not applicable, 2—low need, 3—moderate need, 4—high need).

### 2.5. Procedure

The lead author of this study provided the family caregivers with measuring instruments. The research was carried out in pediatric oncology and hematology departments of clinical hospitals and oncology clinics. The interviews lasted about an hour. The sampling was carried out in 2015–2020.

All family guardians interviewed were invited to participate voluntarily; the objectives of the study were explained to them, and all their concerns about the study were addressed. Family caregivers who agreed to participate signed informed consent forms and responded to measuring instruments individually in one session. The participants suffered no consequences for the withdrawal of consent as indicated on the informed consent sheet. Before collecting the answered instruments, the interviewer checked whether there were unanswered questions. The participant was asked to answer any unanswered questions, preventing missing values from occurring.

The study was in line with the applicable international guidelines of the Helsinki Declaration.

### 2.6. Data Analysis

Data analysis was performed with the statistical package TIBCO Software Inc. (2017). Statistica (data analysis software system), version 13. http://statistica.io (accessed on 14 November 2021). License purchased by the University. Variables were described as frequency (%), range (minimum and maximum), mean (M), and standard deviation (SD). To describe the sample, comparisons of socio-demographic variables were made. The null hypothesis of equality of frequency (sex ratio, place of residence, duration of illness) was tested by the Pearson chi-square test or the two-tailed Fisher’s exact test. The analysis used descriptive statistics and confidence intervals in the assessment of participants’ characteristics, metric and demographic data, and in the assessment of problems. Statistical characteristics of continuous variables were presented in the form of arithmetic means, standard deviations, and medians. Statistical characteristics of step and qualitative variables were presented in the form of numerical and percentage distributions, using the Student’s *t*-test or the Mann–Whitney U test. Correlations were determined using Pearson’s test, while χ^2^ was used for intergroup comparison. Significance was assessed at the level of *p* < 0.05. The repeatability of answers to individual questions was assessed using Cohen’s kappa statistics. Missing data were excluded from all analyses.

## 3. Results

### 3.1. Demographic

All respondents in their legal status were parents of children with cancer. A total of 800 parents completed the questionnaire. Among the 800 people included in the study, women constituted 85% (95% CI: 81–89), and men 15% (95% CI: 14–20). The mean age of the mother was 38.2, SD = 7.25, and that of the father was 41.1, SD = 7.03. Other descriptive statistics identifying the studied group are presented in Table 1.

### 3.2. Needs

The research have shown, that 70% of respondents had moderate or high needs in the field of medical information, 55% of parents reported a moderate or high level of need for help in the psychological or emotional field, and 30% in the financial domain. There were no significant differences in terms of sex, marital status, and parents’ education. However, there were differences between the child’s needs and the duration of the illness. The type and intensity of needs were statistically signifi-cantly related to the duration of illness. Parents of children who were ill for less than 3 years more often reported emotional and information needs, while parents of children who were ill for more than 3 years more often reported somatic and financial needs. The differences were statistically significant (*p* = 0.01). In the studied group, there was a strong positive linear relationship between the needs of the psychological sphere and the number of children (+0.993). The results were significantly higher for families with more than one child (Table 2; Figure 1).

### 3.3. Needs by Domain

The analysis found that 97% of the participants experienced some level of need for one or more items, and 73% of the respondents reported a moderate or high need for one or more items. The most common items were the need for emotional support from relatives (91%), the need for information on the child’s health state and prognosis (83%), and assistance in everyday functioning (77%) (Table 3).

## 4. Discussion

A child’s cancer is a clear example of an abnormal event that reorganizes family life. Taking care of an oncologically ill child is an extremely burdensome task, as it is associated with many additional duties. To meet the new requirements, parents must take on new tasks and take up new functions. It often happens that the disease changes the structure of the family and the relationships between its members. It is necessary to reorganize the current life, and change habits and family roles. In the face of a child’s illness, new needs emerge that can apply to every sphere of life. It is worth noting that parents of sick children usually do not receive sufficient support; therefore, they have to deal with their problems on their own. In the literature, they are referred to as hidden patients, because the multitude of tasks and duties that they have to meet may contribute to the occurrence of malfunctions [22,23,24,25,26].

More and more attention is paid to parents of children suffering from cancer, but most of the studies that have so far been conducted in the population of parents of children suffering from cancer have focused on determining the extent to which parents are exposed to experiencing anxiety, stress, and depressive symptoms [7,21,22,23]. Recent literature shows that parents of children with cancer have many unmet needs, and psychosocial needs are of paramount importance. The basis for starting the research was an attempt to supplement the knowledge in the scope of determining the needs experienced by parents of children with cancer and determine their level of dissatisfaction. These studies have shown that 97% of parents of children with cancer have some level of perceived need and that they have moderate or high needs in various areas.

One of the main needs among parents, as shown by our research, is the need for information. Families require access to appropriate information about the condition of their children to be able to participate in healthcare activities. They must have access to clear and understandable information so that they can make the best decisions about childcare. However, parents of cancer patients often complain of incomplete information provided by the medical team about the condition and treatment of their children [6,27]. Similar results obtained in the studies by Qingying et al. showed that information needs were the first among the unmet needs of parents (M = 4.5; SD = 0.68), mainly including information on the treatment of their child, side effects, and ways of coping with the child’s disease. [4]. In a study by Borjalil et al., four main themes were identified in terms of parents’ health information needs: medical, physical health, healthcare, and family lifestyle information [28]. Adamsi et al. identified the main information needs of the families of cancer patients as treatment, diagnosis, coping with the disease, self-care, types of cancer, hospital care, and rehabilitation [29]. Inman showed that for parents of children with cancer, the health information should include the long-term effects of cancer treatment on survivors, concerns about the physical and psychological consequences, the performance of parents and family members, and access to information about behavioral problems, education, sleep disorders, eating, and communication with others [30]. In addition, Mitchell et al. emphasized the need for parents of children with cancer of all age groups and stages of the disease to receive appropriate medical information from doctors and nurses during diagnosis and treatment [31], and Maree et al. stated the need to receive medical information verbally from the healthcare team [32]. Other studies showed the need for information focusing on how to care for a sick child and information about the child’s condition, allowing them to feel a certain control over the situation [17,33,34].

Emotional needs are the second most common category of needs in the literature. The results of numerous studies confirm that the process of diagnosis and treatment is very burdensome for parents [22,25,26,35,36]. Bruce’s research consistently found that parents had higher rates of post-traumatic stress disorder (PTSD) and post-traumatic stress symptomatology (PTSS) than did their children, suggesting that the experience of parenting a child with cancer may be more traumatic than the actual disease [36]. Moreover, parents of children suffering from cancer have more psychological difficulties than do parents in the general population, and thus they also have higher psychological and emotional needs [22,25,26,35,37]. Our research also showed a moderate or high level of psychological and emotional needs, which were ranked second in terms of the frequency of occurrence, including the need for support in this regard. In their research, Qingying et al. presented the conclusion that parents of children with cancer in China expressed the highest needs for informational and emotional support, and the emotional needs of parents mentioned were fear, worries, sadness, and other concerns (emotional needs M = 4.12, SD = 0.94; psychosocial needs M = 3.94, SD = 0.87) [4]. According to other authors, the most important emotional need was the feeling of uncertainty about the child’s cancer, cure, diagnosis, treatment, symptom management, and the future [34,38,39,40]. Therefore, the need for support among parents of children with a life-threatening disease seems to be particularly significant. Mitchell et al. identified the need for emotional support among 45% of parents, pointing not only to key healthcare workers, but also to informal support [31]. Loghmani et al. mentioned emotional needs among families, such as psychological support, empathy, and mutual understanding [41]. As Tomlinson and Mitchell point out, support may be of the greatest importance in moments of extreme crisis. When planning therapeutic interventions, it should be taken into account that the impact of supporting the immediate environment and launching effective coping strategies can significantly improve the functioning of the parent, and thus support the process of their adaptation. These variables have a protective function and give hope for a relatively healthy adaptation. It also means that a child’s illness does not always have to be associated with a long-term emotional crisis [42]. Moreover, there are scientific reports showing that this experience may lead to many positive changes in parents, which may occur as a result of success in coping with the demands posed by the child’s disease [7,43].

In the authors’ research, the third place among the needs was taken by financial needs. They may result from high costs of treatment, medical supplies, or health support, exceeding the level of income of a given family. It is not uncommon for one of the parents to resign from work to take care of a sick child. Usually, it is a mother who, to meet the requirements posed by the child’s illness, leaves her job or limits the number of hours worked, which translates into a decrease in the family’s income [44]. In their research, Shamsaei et al. mentioned financial needs related to care and treatment [45]. The occurrence of financial needs and highly valued support in this area has been demonstrated in numerous studies [46,47,48,49,50,51]. In contrast, Qingying et al. showed that as long as the family was assured that the child would receive treatment, financial needs were less of a priority than were other logistical problems [4].

As shown by data in other studies, parents, as persons directly involved in the care of a child and the process of their treatment, may experience many difficulties. They are required to fulfil their daily duties, burdening them with a huge number of tasks and roles. They have to function in two different environments, and in each of them, they have specific tasks to perform, which can lead to overload and fatigue, both physical and mental [22]. Some reports indicate the need to manage daily activities, as well as the need for practical help in cooking, receiving mail, and caring for siblings, giving respite from care and achieving a sense of normality [47,50,52,53,54,55,56]. The physical needs cited in the literature describe the parents’ needs for help in understanding and coping with the physical symptoms of the child’s disease, but also include looking for support for their physical symptoms such as difficulty sleeping, fatigue, loss of appetite, weight loss, dizziness, and headaches [39,50,56,57,58]. In our research, parents indicated the need for help in coping with their own symptoms and help in everyday functioning. A very small group indicated the need for support in intimate life, but no reference to this need was found in the literature.

Spiritual needs were identified by a smaller group of parents. The parents indicated the need for spiritual support and the need to talk about transience. According to other studies, regardless of their previous beliefs, parents sought meaning for their experience and possible explanations for their child’s illness as well as spiritual support from monks and in religious practices [33,50,57,58,59].

This study provides an initial understanding of the unique needs of parents of children with cancer. The presented data may be useful in clinical practice, as they draw the attention of specialists to the difficulties of parents and allow them to identify their needs, provide them with appropriate, tailored care, and above all, to meet the unmet needs of parents. The presented results may contribute to the development of standards of work with parents. Data showing that the stage of a child’s treatment and the time since diagnosis are significant variables in the adaptation process seem particularly important to practitioners. Limitations of this study are the small sample size, short evaluation time, and a limited number of items for assessment of needs. Further research aimed at increasing the sample size as well as strengthening elements that relate to each category of needs, including the existential aspects and the suffering of the child, is recommended.

## 5. Conclusions

(1)Parents of children with cancer experience a high level of needs, especially psychological, emotional, and information. These data suggest that the existing healthcare system does not meet the needs of parents of sick children. The results show the need to investigate the mechanisms by which healthcare providers can use the healthcare system to identify and meet needs.(2)Parent-oriented care should be included in pediatric oncology departments. It is possible, thanks to the development and application of strategies, to understand the needs of parents, respond positively to needs, and provide effective care, all to reduce the needs of this vulnerable group and thus minimize the risks of not being satisfied.

## Figures and Tables

**Figure 1 children-09-00144-f001:**
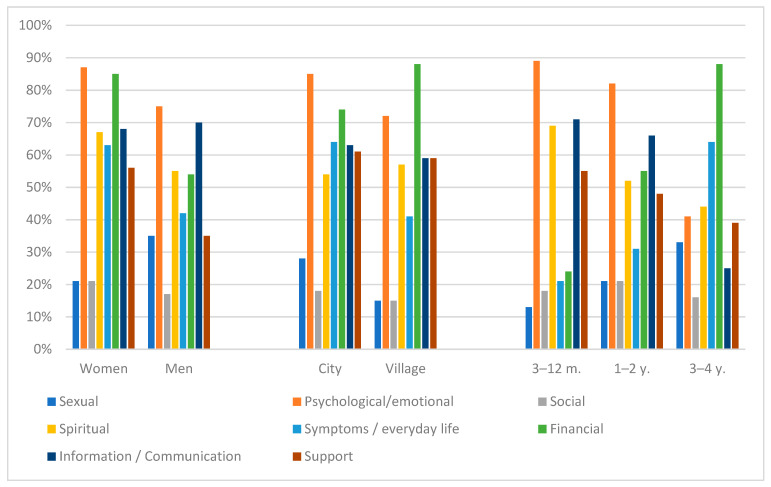
Moderate and high parents’ needs: sex, place of residence, time of illness of the child.

**Table 1 children-09-00144-t001:** Descriptive statistics of the examined group of patients.

Demographic Information	Total *n* = 800	*p*
**Characteristics % (*n*)**		
Sex		
women	85% (680)	0.01
men	15% (120)	
The age of the study group		
SD	44.1 (7.76)	0.12
95% CI	[26; 57]	
The age of women		
± standard deviation	38.2 ± 7.25	0.21
scope	[26; 57]	
median	38	
95% CI	[39.8; 41.8]	
The age of men		
± standard deviation	41.1 ± 7.03	0.19
scope	[26; 57]	
median	41	
95% CI	[39.8; 41.8]	
Place of residence		
city	68% (544)	0.21
village	32% (256)	
Financial situation		
very good	1% (8)	0.01
good	8% (64)	
average	68% (544)	
bad	10% (80)	
very bad	13% (104)	
Age groups		
20–29	3% (24)	0.01
30–40	35% (280)	
41–50	37% (296)	
51–60	25% (200)	
Education of the study group		
higher education	47% (378)	0.01
secondary education	35% (276)	
vocational education	18% (146)	
primary education	0% (0)	
Marital status		
married	74% (592)	0.62
widowed	3% (24)	
unmarried	23% (184)	
Source of income		
professionally active	76% (608)	0.59
annuity	15% (120)	
benefit	9% (72)	
Type of cancer in the family		
leukemia	54% (432)	0.07
brain tumors	19% (152)	
solid tumors	27% (216)	
Age of children with cancer		
up to 5 years	22% (176)	0.19
5–10 years	51% (408)	
11–18 years	27% (216)	
Number of children		
one child	45% (360)	0.71
two children	41% (328)	
three children	10% (80)	
four children	4% (32)	
Times of illness		
3–12 m	43% (344)	0.01
1–2 y	37% (296)	
3–4 y	20% (160)	

**Table 2 children-09-00144-t002:** Moderate and high parents’ needs.

Needs	Moderate/High Needs % (*n*)		Moderate/High Needs % (*n*)		Moderate/High Needs % (*n*)			*p*
	Sex		Place of Residence		Times of Illness			
	Women	Men	City	Village	3–12 Months	1–2 Years	3–4 Years	
Support								
support in dealing with depression	31% (211)	35% (42)	37% (201)	24% (61)	38% (131)	24% (71)	21% (34)	0.91
support in dealing with frustration	56% (381)	31% (37)	61% (332)	59% (151)	55% (189)	48% (142)	39% (62)	0.55
emotional support from loved ones	29% (197)	18% (22)	22% (120)	31% (79)	33% (113)	27% (80)	24% (38)	0.44
psychological support from support groups	20% (136)	15% (18)	18% (98)	17% (43)	18% (62)	7% (21)	11% (18)	0.55
psychological support from medical staff	45% (306)	30% (36)	41% (223)	28% (72)	44% (151)	35% (104)	25% (40)	0.91
Information/Communication								
need information about health status	66% (449)	70% (84)	58% (315)	54% (138)	71% (244)	55% (163)	19% (30)	0.01
need information about treatment	58% (394)	69% (83)	61% (332)	52% (133)	56% (193)	30% (89)	14% (22)	0.01
the need for education about illness from medical staff	68% (462)	59% (71)	63% (343)	59% (151)	70% (241)	66% (195)	25% (40)	0.01
the need for information about the factors that may influence the course of the neoplastic disease	51% (347)	48% (58)	53% (288)	42% (107)	46% (158)	26% (77)	7% (11)	0.41
need information about health opportunities	44% (299)	53% (64)	45% (245)	55% (141)	34% (117)	15% (44)	18% (29)	0.01
the need for complete information on the results of medical examinations	52% (354)	61% (73)	45% (245)	55% (141)	50% (172)	48% (142)	22% (35)	0.01
Financial								
support in everyday life finances	85% (578)	54% (65)	74% (402)	88% (225)	22% (76)	55% (163)	88% (141)	0.01
support in treatment costs	25% (170)	20% (24)	21% (114)	25% (64)	24%( 82)	48% (142)	74% (118)	0.01
Symptoms/everyday life								
help in dealing with symptoms	63% (428)	42% (50)	64% (348)	41% (105)	15% (52)	29% (86)	64% (102)	0.01
help in everyday functioning	32% (218)	21% (25)	30% (163)	25% (64)	21% (72)	31% (92)	35% (56)	0.01
Spiritual								
changing priorities	39% (265)	21% (25)	40% (218)	25% (64)	33% (113)	34% (101)	29% (46)	0.91
help in dealing with the problem of dying	67% (456)	55% (66)	54% (294)	57% (146)	69% (237)	52% (154)	44% (70)	0.71
Social								
ability to express feelings	21% (143)	17% (20)	12% (65)	15% (38)	15% (52)	10% (30)	13% (21)	0.74
planning the future	19% (129)	12% (14)	18% (98)	11% (28)	18% (62)	21% (62)	16% (25)	0.91
support in the functioning of the family	14% (95)	10% (12)	14% (76)	8% (20)	15% (52)	7% (21)	8% (13)	0.55
Psychological/emotional								
help in dealing with worries about prognosis	84% (571)	75% (90)	81% (441)	72% (184)	88% (303)	74% (219)	31% (50)	0.01
help in dealing with frustration	87% (592)	61% (73)	85% (462)	71% (182)	89% (306)	82% (243)	41% (66)	0.01
help in dealing with fears for the family	58% (394)	46% (55)	55% (299)	60% (154)	55% (189)	41% (121)	20% (32)	0.01
Sexual								
support in intimate life	21% (143)	35% (42)	28% (152)	15% (38)	13% (45)	21% (62)	33% (53)	0.19

**Table 3 children-09-00144-t003:** Needs of parents by domain.

Needs	Level %		
	Low	Moderate	High
support in dealing with depression	34%	30%	36%
support in dealing with frustration	14%	58%	28%
emotional support from loved ones	0%	9%	91%
psychological support from support groups	69%	20%	11%
psychological support from medical staff	26%	41%	33%
psychologist support	28%	41%	31%
emotional support of the clergyman	74%	11%	15%
talks about passing and death	34%	28%	38%
the need for information about the state of health	0%	17%	83%
need information about treatment	2%	28%	70%
the need for education about the disease on the part of medical personnel	13%	38%	49%
the need for information on prognosis	31%	41%	28%
the need for information about health opportunities	11%	38%	51%
support in the finances of everyday life	0%	83%	17%
support in treatment costs	58%	25%	17%
help in dealing with symptoms	20%	65%	15%
help in everyday functioning	07%	23%	77%
changing priorities in life	40%	42%	18%
help in dealing with the problem of dying	9%	40%	51%
the ability to express feelings	60%	25%	15%
planning for the future	73%	17%	10%
support in the functioning of the family	74%	14%	12%
support in intimate life	43%	39%	18%
striving to achieve life goals and desires	75%	15%	10%
learning and personal development	82%	10%	8%
recognition from other people	68%	21%	11%
maintaining social contacts	66%	19%	15%
need for faith and religious needs	64%	21%	15%
understanding and love	21%	31%	48%
being able to be independent and self-sufficient	32%	29%	39%
care/medical visits	9%	32%	59%
nursing care/visits	6%	33%	61%
respect and subjective treatment	0%	28%	72%

## Data Availability

Data available on request due to restrictions of privacy and ethics.

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
