# Peer review of "The Needs of Parents of Children Suffering from Cancer—Continuation of Research"

_children, 2022, doi:10.3390/children9020144_

Round 1

Reviewer 1 Report

The Author shows that parents of children with cancer have
high levels of unmet needs, including psychosocial, emotional, physical, informational, financial, educational, and spiritual needs. 
The study was conducted between 2015 and 2020 and examined the prevalence of unmet needs among parents of children with cancer from a region of southern Poland.  It was found that as high as 97% of parents felt some level of need for one or more supports, and 73% of respondents reported moderate or high need for one or more supports.
70% of respondents had moderate or high needs for medical information,
55% of parents reported a moderate or high need for help with psychological or emotional, and 30% in the financial area. The results of the study showed that parents of children with cancer experience high levels of needs, especially psychological, emotional, and informational needs. These data suggest that the existing health care system is not meeting the needs of parents of sick children.

Author Response

Dear Reviewer,

Thank you for any comments from the reviewer.

Sincerely

Anna Lewandowska

Reviewer 2 Report

Manuscript ID: children-1532501: The needs of parents of children suffering from cancer-continuation of research

Thank you for the opportunity to review the manuscript: The needs of parents of children suffering from cancer - continuation of research.

I appreciate the amount of work the authors have put into the manuscript.

  Overall, the manuscript is well written and easy to follow. However, following a careful reading of the manuscript, I have some comments:

Thank you for the opportunity to review the manuscript: The needs of parents of children suffering from cancer - continuation of research.

I appreciate the amount of work the authors have put into the manuscript.

  Overall, the manuscript is well written and easy to follow. However, following a careful reading of the manuscript, I have some comments:

Title:

It is adequate, clear, specific and concrete, as established by the criteria of the MDPI journal Children.

ABSTRACT:

It is adequate, it has been well written, it is concrete and if it integrates the elements: objective, method and findings. As established by the criteria of the journal Children of MDPI. Aspects that contribute and facilitate the reading and understanding of the manuscript.

INTRODUCTION:

  1. In the Introduction section, the manuscript must describe a technically sound piece of scientific research with data that supports the conclusions. I suggest that the authors work on the manuscript until they show scientific evidence that supports the research question from the available empirical literature.

  1. The first paragraphs seem correct to me, however, I suggest to the authors that in addition to reporting the empirical evidence available about the needs of parents of parents of children with cancer, the epidemiological evidence of the problem, incidence and prevalence of this problem be reported. complex problem from the international scientific literature, as it should be in an article that will be published in the MDPI journal Children.

In addition, a fundamental aspect for the closing of the paragraphs of the introductory section, in addition to the fact that the authors have described the factors that explain the needs of parents of parents of children with cancer, it is important that the authors also present empirical findings of the resilience processes in diverse contexts, populations and cultures; Because currently the focus on meeting the needs of parents has focused on developing positive adaptation processes to overcome adversity, risk and vulnerability in families and patients with cancer. To do this, I suggest reviewing and citing the following research results:

TO). https://hqlo.biomedcentral.com/track/pdf/10.1186/s12955-017-0817-3

B). https://bmcpublichealth.biomedcentral.com/track/pdf/10.1186/s12889-019-7512-8

C). https://res.mdpi.com/d_attachment/ijerph/ijerph-18-00748/article_deploy/ijerph-18-00748-v3.pdf

  1. These paragraphs of the introduction seem adequate to me, however, it is necessary to make a link between the needs of parents of parents of children with cancer, and their impact with the dissatisfaction of needs in the family caregivers of children with cancer. Considering that this manuscript is focused on the needs of parents of parents of children with cancer, I suggest that the authors write a paragraph at the beginning of the first paragraphs of the introduction to contextualize the impact of chronic disease in families, I suggest review and citation of the following articles:
  2. a) The caregiver's burden https://bpsmedicine.biomedcentral.com/track/pdf/10.1186/s13030-019-0147-2
  3. b) Anxiety in family caregivers in contexts of significant adversity:

  1. I suggest to the authors that within the introductory paragraphs they include the definition of family caregiver in contexts of chronic disease, since the manuscript in its current form does not theoretically and empirically place the family caregiver, despite being the population in which The unmet needs of parents of children with cancer were obtained. Therefore, the authors do need to include a definition of family caregiver published in the international literature, “In the research literature, a family caregiver is defined as a person who has a significant emotional bond with the patient; this caregiver is a family member who is a part of the patient's family life cycle, offers emotional-expressive, instrumental, and tangible support, and provides assistance and comprehensive care during the chronic illness, acute illness, or disability of a child, adult, or elderly person ”, here I share a link to the article for review and citation: https://journals.plos.org/plosone/article/file?id=10.1371/journal.pone.0206917&type=printable

  1. Likewise, in the paragraphs of the introductory section, in addition to the psychosocial factors that the authors have mentioned related to the satisfaction of parents' needs; It is important that they be able to add more empirical evidence focused on some fundamental aspects that characterize the factors that influence risk associated with the impact of care and the consequences of care in families of children with cancer and chronic diseases. These suggested articles will allow to support the Discussion section of the present manuscript from the point of view of the dissatisfaction of needs. For this, I suggest the review and citation of the following research results that show empirical findings and that may contribute to support the present manuscript:
  2. a) The quality of life in parents of children with chronic diseases

b). The coping and health variable in family caregivers facing the disease https://hqlo.biomedcentral.com/track/pdf/10.1186/s12955-020-01357-5

c). And the anxiety variable in family caregivers of children with chronic diseases.

https://bpsmedicine.biomedcentral.com/track/pdf/10.1186/s13030-018-0139-7

d). Social support networks in contexts of chronic disease

and). And how does the caregiver profile influence anxiety in primary caregivers of children with disabilities?

https://www.ncbi.nlm.nih.gov/pmc/articles/PMC7583305/pdf/13030_2020_Article_201.pdf

  1. I suggest that the authors make a clear and concrete statement of the problem, which guides the idea and the research question, so that they can make a theoretical, practical, social and methodological justification of the present manuscript.

  1. I suggest that the authors fully include each of the 6 previous aspects that I have indicated for the introduction section. Doing this can successfully contribute to a better understanding of the Background / Introduction found in the literature review, leading the reader in a systematic way to the method of the present study.

METHODS

  1. In the method section the experiments must have been conducted rigorously, with appropriate controls, replication, and sample sizes.

I suggest that the authors resolve each of these aspects, to improve the final version of the manuscript, as it is not yet clear within the manuscript.

  1. I suggest that at the beginning of the method section, describe the type of study and the scope of the investigation; the procedure for selecting the participants and defining the type of probability or non-probability sampling through which the authors collected data.

  1. In the Study design and Participants section, I suggest clarifying more precisely and unambiguously the design and the total number of participants.

  1. In the measurement instruments section, the authors have reported the psychometric properties of the instrument; However, I suggest reporting the number of items and the factors, domains or dimensions of the scale, I suggest that they describe the Cronbach's alpha coefficients of the measurement instrument, and I suggest that the authors report the internal consistency, and the total variance explained of that instrument. Finally, I suggest that the authors incorporate the citation and the reference from which the instrument is based.

  1. The data analysis seems adequate to me. Well, the authors perform statistical tests of association and differences. Aspects that are very important when studying the burden of the caregiver. I suggest adding the name, version and country of the Statistical Package with which the authors developed the data analyzes. Also, that they describe the tests used to report the effect size, as that is important.

RESULTS:

  1. Conclusions will be properly drawn based on the data presented.

  1. The results have been very well worked.

  1. I suggest expanding the presentation of the results, considering the objective of the study, the research variables, and the findings from the statistical tests used.

DISCUSSION

  1. The discussion does provide elements and theoretical and practical implications for research on the needs of parents of a child with cancer and the level of dissatisfaction.

CONCLUSION

I suggest that the authors rewrite the conclusions of this research, based on the findings of the present study, and its theoretical and methodological implications of the outcome variables, and for future research in the field of chronic diseases and their impact on the carer.

Author Response

Dear Reviewer,

Thank you for any comments from the reviewer. Changes were made as suggested by the reviewer:

  • introduction: more data has been entered, the introduction has been clarified, and the articles have been cited
  • methods added information about the research method, inclusion criteria, study participants, procedure, statistical analysis
  • results: charts added.

The article did not include the quality of life and risk factors, as it was not the content of the study. Taking the reviewer's tips into account, future studies will focus more on anxiety in caregivers and factors that influence the needs and anxiety levels to assess dependence. The author will be happy to use the experience of researchers in the indicated publications.

Sincerely

Anna Lewandowska
